# Improved Accuracy of a Single-Slit Digital Sun Sensor Design for CubeSat Application Using Sub-Pixel Interpolation

**DOI:** 10.3390/s21041472

**Published:** 2021-02-20

**Authors:** Fuat Kaan Diriker, Alexander Frias, Ki Hwan Keum, Regina S. K. Lee

**Affiliations:** Department of ESSE, York University, Toronto, ON M3J 1P3, Canada; kdiriker@my.yorku.ca (F.K.D.); friasal@yorku.ca (A.F.); pkeum@yorku.ca (K.H.K.)

**Keywords:** digital sun sensor, single-slit design, sub-pixel interpolation, CubeSat

## Abstract

In recent years, we have seen significant improvements in the digital sun sensor (DSS) design enabled by advanced micro-systems fabrication and optical sensing technologies. In this paper, we propose a simple single-slit DSS concept with improved accuracy using sub-pixel interpolation. In considering the DSS design, we focused on several characteristics of the sun sensor, including field-of-view, sensor accuracy, complexity, and computational requirements. First, the optimal mask design was determined based on the simple geometry of the slit size, mask height and pixel width. Then, a two-step pixel read-out algorithm was implemented for sub-pixel level interpolation to determine the illumination ratio using 1-, 2-, 4- and 8-bit readouts. Lastly, the improved DSS was integrated with typical CubeSat, commercial-grade attitude sensors suite and a simple TRIAD method to determine the attitude of a CubeSat in LEO. Compared to standard 1-bit read-out mode (0.32 deg RMSE), 8-bit DSS achieves better than 0.01 deg RMSE. In a CubeSat scenario, improvements in attitude knowledge and control accuracy are marginal when using TRIAD, due to the significantly lower accuracy in other CubeSat-scale sensors (magnetometer, for example).

## 1. Introduction

The attitude control system (ACS) plays a critical role in spacecraft design and is vital to the success of a space mission. Generally, the ACS consists of various sensors, several actuators, and complex control algorithms. With recent technological advancements in the microelectromechanical systems (MEMS) and micro-sensor sectors, attitude sensors like the digital sun sensor (DSS) continue to improve with compact packaging, small mass and power consumption, redundancy, and reliability—especially benefiting small satellites. However, much more improvement is still needed and can be attained in accuracy and computational requirements. Recognizing the need for improvement in DSS performance, Wei et al. [1] proposed a wireless DSS with 100-degree field-of-view (FOV) and 0.01-degree accuracy (1-σ) using sub-pixel interpolation of CCD imagers. Alvi et al. [2], improved the DSS accuracy using sub-pixel evaluation and centroiding with simple and inexpensive commercial grade components. Similarly, the commercial MiniDSS from TNO, which achieved an improved accuracy in the order of 10^−2^ degree using a 15-bit readout that yielded 1/64-pixel accuracy with just a mass of 72 grams [3].

The design optimization of optical sensors (such as star trackers and digital sun sensors) using centroiding to achieve sub-pixel level accuracy is a well-established method. In the context of star tracker image processing, centroiding refers to the process of locating the star center in a star image frame [4]. Together with thresholding, centroiding algorithm has been studied by numerous authors to improve the imager accuracy to sub-pixel accuracy. The most basic centroiding technique is intensity-weighted centroids, also referred to as center of gravity or momentum method. In its simplest forms, the centroid coordinates $\left(x_{c},y_{c}\right)$ are computed by dividing the sum of pixel location, $X_{ij}$, multiplied by the intensity $I_{ij}$ by the total intensity, as given by equation $\left(x_{c},y_{c}\right)=\left(\frac{\sum_{i}I_{i}x_{i}}{\sum_{i}Ii},\frac{\sum_{i}I_{i}y_{i}}{\sum_{i}Ii}\right)$. Based on a series of simulation studies conducted by Li et al. [5], a 0.5-pixel accuracy was achieved using this simple centroiding algorithm. In DSS design, as opposed to a slit design, centroiding is used with pin-point mask, where a multi-pixel image of a ‘sun spot’ is expected.

In the authors’ previous paper [6], an array-based multi-slit aperture DSS design was proposed using two simple photodetector arrays. While the proposed design met most of the CubeSat attitude knowledge requirements (approx. 1-degree accuracy), further improvements are still needed to achieve fine-pointing operation. Sub-pixel interpolation has also been implemented to extract the sun vector from a non-ideal reading due to glare [7]. The glare, causing a “black sun” phenomenon due to over-saturation of a pixel, was compensated for using sub-pixel centroid detection, and an accuracy of 0.05 degrees was obtained by fusing measurements from three sun sensors.

Other methods of improving the sun sensor have also been explored, for example, by using a different aperture mask or sensor construction. A single linear array CCD combined with a V-shaped aperture mask was proposed by Fan et al. to widen the FOV while simplifying the required algorithm. The experimental FOV was 65 degrees in both axes, while achieving a 0.1-degree accuracy within the FOV [8]. In another paper, improvement of the sun sensor through usage of a different wafer material for the solar cell array has been explored by Hales et al. [9], where a two-axis micro-opto-electro-mechanical system (MOEMS) sun sensor was implemented using a silicon-on-insulator (SOI) wafer, yielding a FOV of 70 degrees in all axes with a theoretical resolution of 0.07 degrees. This MOEMS sun sensor has been implemented and flown in DTUsat-1 and DTUsat-2 [10]. However, these improvements come with a requirement of a major physical modification, which may not be feasible for a matured sun sensor design. Thus, in this paper, we propose a single-slit DSS designed primarily for CubeSat applications using sub-pixel level interpretation to enhance accuracy limitation. Furthermore, the characterization results from the prototype sensor are expected to be implemented on University of Manitoba’s upcoming IRIS mission (previously known as ManitobaSat-1) [11]. The 3U spacecraft used in the IRIS mission aims to study the space weathering of geological and lunar samples over time.

In determining the optimal design for the proposed DSS, several criteria were considered, namely: readout options, FOV, sensor accuracy, complexity, and computational requirements for sub-pixel interpolation. Compared to the authors’ previous design, the current IRIS DSS featured a simple interface, an improved accuracy and a stand-alone package for convenient mechanical mounting, all within a 3 cm^2^ footprint [6]. Lastly, this paper is organized in the following manner. An overview of the literature and research is provided in Section 1. Section 2 provides an overview of the DSS concept, mechanical design, sub-pixel interpolation implementation, and hardware-in-the-loop simulation. Section 3 discussed several key results of the study, including performance at different readout modes at different scenarios. Finally, Section 4 gives a brief summary of the research and future works.

## 2. Materials and Methods

### 2.1. Description

As seen in Figure 1, the DSS is a dual-axis linear array-based sensor which consists of two orthogonal linear photodiode arrays (Melexis MLX75306 shown), decoupling capacitors, and custom-designed aperture masks to create an incidence pattern. Key performance specifications of the sun sensor are summarized in Table 1.

The proposed design in this study is a variation of the five-slit design that was implemented on the DESCENT CubeSat mission [12]. The five-slit design reduced the potential readout errors by creating the incidence pattern five times, ultimately creating redundancy in the sensor design. To further enhance the reliability and redundancy in the ACS design, a pair of the DSSs will be installed on IRIS spacecraft in lieu of a multi-slit design. The readout error check is achieved by comparing outputs from two sensor units. The single-slit design considerably reduces the computational requirements for the microcontroller, since the pattern is processed only once. The operational concept of the DSS is based on a simple trigonometry; as seen in Figure 2, the sun incidence angle, θ, is calculated from the lit pixel pattern on the photo-array from the photo-mask (shown as the lightly shaded region with thickness, *t*). The key parameters in the incidence calculation are: thickness of the mask ceiling, *t*; the height between photodiode array surface and inner surface of mask ceiling, *h*; and incidence length with respect to the reference pixel, *d*. The parameters $x_{i}$ and $x_{f}$ are the beginning and the ending positions of the incidence pattern caused by the incoming light. Here, *l* is the shadow length induced by the thickness of the mask and incidence angle. Usage of these parameters is described in Section 2.3.

Here, the reference pixel is defined as the pixel that is directly under the slit of the mask; as shown in red in Figure 2, this corresponds to the 62nd pixel. Given the size and mass of the sensor, and availability constraints of photo-arrays, design parameters were carefully considered, with the goal of maximizing the FOV while maintaining the desired target accuracy, as outlined in the next section.

### 2.2. Mask Mechanical Design

Several parameters for the mask design were determined using MATLAB to achieve the desired accuracy and FOV. The key mechanical parameters affecting the FOV include: slit width, ceiling height, and mask ceiling thickness. Furthermore, without compromising cost, the parameters were constrained to a reference distance of 0.5 mm due to the manufacturability limitations. Figure 3 illustrates the plot of attainable FOV AT 5-m. In addition, the offset of the incidence pattern from the reference pixel is also considered, since the number of pixels on the photodiode array is finite. Several design iterations were considered, however, the physical parameters of the final designs chosen are presented in Table 2.

### 2.3. Sub-Pixel Interpolation in Sun Sensor Readout

The sun sensor previously modeled computationally in Bolshakov et al. [6] only concentrated to the 1-bit readout mode of the DSS. However, this paper extends the computational model to account for the 2(1.5)-bit, 4-bit and 8-bit readout modes. Based on the trigonometric relationship outlined in Section 2.1 and in Figure 2, the incidence patterns expected to form on each individual photodiode array were simulated. To simulate these patterns, the first pixel starting from the left-hand side that would be illuminated is identified using the incidence length, *d*, as follows:
(1)$$d=\frac{h\tan\theta }{\text{pixel size}}.$$
Next, the calculated value is subtracted from the reference pixel identified in Section 2.1, which is pixel number 62. The result is the first pixel from the left, denoted as $x_{i}$ in Figure 2. To illustrate this, for a value of θ = 30 deg, the incidence length is d = 23.787 pixels. Subtracting this value from the reference pixel yields the starting position of the incidence pattern at $x_{i}$ = 38.213 pixels. It is worth noting that since pixel locations are defined as whole numbers, the calculated result would mean that one pixel is not completely illuminated. If incidence starts from pixel 38.213, the 38th pixel would not be illuminated and the incidence would start from the 39th pixel. Furthermore, only about 78 percent of the pixel 39 would be illuminated. To complete the pattern, the length of the incidence, or in other words, the number of illuminated pixels were calculated using the following trigonometric relationship:
(2)$$l=\frac{t\tan\theta }{\text{pixel size}}.$$


When the incoming light is along the normal vector of the photodiode array, that is θ = 0 deg, the shadow length must also be equal to 0. The unobstructed incidence length generated in this configuration is equal to the slit width, w = 1.00 mm. A pixel pitch size of 50 m would correspond to 20 illuminated pixels. However, for the case where θ = 30 deg, the shadow length is computed as l = 8.660 pixels. Subtracting this value from the unobstructed incidence length yields 11.340 pixels. This means that 11.340 pixels are illuminated starting from $x_{i}$. Adding the length of illuminated pixels to $x_{i}$ would yield the location of the end pixel, as seen in Figure 2. For the case θ = 30 deg, $x_{f}$ = 49.553 pixels. Contrasted to the computation of the first pixel, pixel 49 would be fully illuminated, and about 55 percent of 50th pixel is illuminated.

Repeating this two-step strategy for the entire FOV range between −52.5 to +52.5 degrees, at intervals of 0.5 deg, yields the angle-pixel performance shown in Figure 4, with Figure 5 showing a zoomed portion to illustrate the where grey-scale values were used to indicate different illumination radius. The different shades are a result of converting coverage ratio to RGB values, where a 100% and 10% illuminated pixel would have RGB values of (1, 1, 1) and (0.1, 0.1, 0.1), respectively.

The MLX75306 photodiode array used in this study consists of 144 pixels; except for the sanity pixels (1st and 144th pixel positions), each pixel can readout incidence intensity using a dedicated analog-to-digital converter (ADC) in either 1-, 2(1.5)-, 4- or 8-bit readout modes. Therefore, if a pixel is completely illuminated, the pixel would return values 1, 2, 15, or 255 depending on the readout mode. This, however, introduces a discretization error during hardware readouts. To illustrate using a 1-bit readout mode, a 65%-illuminated pixel would return a decimal value of 1, while a 45%-illuminated pixel would return a decimal value of 0. In contrast to the 8-bit readout mode, it would return decimal values of 166 and 115, for 60%- and 40%-illuminated pixels, respectively. Again, this introduces a discretization error since the readout 166 actually corresponds to 65.098% and the readout 144 corresponds to 45.098% illumination. This discretization error was introduced to the computational photodiode array model outlined above by dividing each pixel to multiple sub-pixels, and depending on the readout mode, the illumination ratio was multiplied by the maximum possible output of the specified readout mode and rounded to the nearest integer. For the 8-bit mode with 60% illumination, the computational model would have returned a readout of 165.75, a decimal value that cannot be returned by the hardware. In which case, the computer model would mimic the ADC and generate the output 166, therefore introducing the discretization error. This discretization error is expected to be more outstanding in the 1- and 2(1.5)- bit output modes.

It is worth noting that the photodiode array has a limitation on the 2-bit readout mode, where consequently, it is referred to as the 1.5-bit mode. Normally, a completely illuminated pixel (above 66.6% illumination threshold) in the 2-bit mode would be expected to return a binary value of 11, however, in order to allow for the device to calculate the average byte correctly, the output 11 is not valid, and would therefore return a binary value of 10 for a fully illuminated pixel. The average byte contains the average value of all active bytes that are read out. The generated output simulates the expected output format from the photodiode arrays and particularly useful to accurately simulate the readout format from the sensor, as this enables the same simulation to be easily adapted to the hardware-in-the-loop testing outlined in Section 2.4. An angle determination algorithm was developed to determine the start position of the incidence, and using the distance *d* from the pixel readout, the angle θ can be estimated.

### 2.4. Hardware-in-the-Loop (HWIL) Simulations

As previously mentioned in Section 2.3, the incidence pattern generation algorithm is expanded to accurately simulate the readout format of the sun sensor. The algorithm is considered in the hardware-in-the-loop simulations to determine the sensor refresh rate and validate the angle determination software. Two different hardware-in-the-loop simulation configurations were considered; the first configuration aims to validate the payload computer algorithm, and to measure the computation time for different angles. The second configuration aims to determine the refresh rate of the sun sensor and payload computer pair.

The first configuration is shown in Figure 6, where the angle detection algorithm is flashed into an ATmega328PB 8-bit AVR microcontroller unit (MCU). For this setup, the universal asynchronous receiver/transmitter (UART) interface and one general-purpose input/output (GPIO) of the MCU were enabled. The UART interface was used to connect the MCU to the PC. In this configuration, the PC acts as the sun sensor and a raw bitstream is outputted to emulate the output of the sun sensor for different angles.

The flow of this test can be summarized as follows: The MCU initially requests data from the PC, as it would during flight operations, and receives the emulated sun sensor data. From the received data, the angle is calculated and transmitted back to the PC, where the relative angle error can be determined. The GPIO pin is designated to act as a flag to determine timing. The pin is programmed to drive high after transmission is complete and to drive low after the angle is calculated. These data are later used to determine the angle computation time. The results of the accuracy calculations can be seen in Table 5 in Section 3. It can be noted that computational time between different angle tests is negligible using this configuration.

The system architecture of the second configuration can be seen in Figure 7, with the corresponding test flow for both configurations given in Figure 8. This configuration aims to determine the sensor refresh rate, i.e., the shortest time between two consecutive measurements, and validate the SPI driver software between the MCU and the sun sensor. While the first setup logged the angle computation time, this only accounted for the time it took for the MCU to calculate the angle, and does not consider the readout time from the sun sensor. However, in second configuration the sun sensor is also included in the measurement. To represent the sun sensor, one photodiode array was added to the test configuration. This array was identical to the arrays on the sun sensor (MLX75306) and is interfaced to the MCU through the SPI bus. The SPI bus is also connected to the logic analyzer to accurately determine the readout time of the IC. This test showed that the computation times of different angles were almost identical.

Since the angular accuracy tests were executed during the first test configuration, instead of sending emulated data from the PC, emulated sensor output was hard-coded to the MCU. The flow of this test was as follows; The MCU commanded the X-axis photodiode array to acquire and transmit data while pulling the GPIO flag low. The MCU then requested data from the Y-axis photodiode array. The MCU calculated the X-axis sun angle while waiting for the sensor to return the Y-axis data output. Once the Y-axis output was received, the MCU calculated the angle again. The GPIO was pulled high to indicate that the operation was completed, and the MCU transmitted both angles back to the PC. This test was repeated for the 1-, 1.5-, 4- and 8-bit readout modes.

The results of the refresh rate tests are presented in Table 3. The angle data length and complete data length are the properties of the photodiode array, and differ depending on the readout mode. The total time refers to the cumulative time between the PC requesting data from the MCU and PC receiving the requested data. Since the interface between the PC and the MCU will be replaced with SPI on the spacecraft, the UART transmission times were omitted. Instead, the transmission time of a data packet of the same length was calculated for each readout mode, assuming an SPI clock speed of 2 MHz. This test showed that the proposed algorithm and data acquisition method can theoretically result in refresh rates higher by an order of magnitude than reported by Wei [1], and potentially better than some commercially available sun sensors for CubeSats such as: NewSpace Systems (NSS) NFSS-411/NCSS-SA05, BiSon64-ET-B (FM 700S00401), and SolarMEMS nanoSSOC-A60.

Along with an HWIL simulation, a day-in-the-life (DITL) orbit simulation is conducted to examine the performance of the sun sensor in a practical setting. While numerous researchers [13,14,15] have reported on the attitude control accuracy of CubeSat-class spacecraft ranging from arc-minute level precision to 30-degree accuracy, knowledge error is often omitted. Furthermore, on-orbit performance of attitude determination system is not easily assessed; only the estimated control accuracy is reported, with knowledge error being part of the uncertainty. By measuring this error through a DITL simulation, this paper aims to validate the design of the proposed DSS with advanced algorithm for sub-pixel level accuracy and allow practical improvements. As a baseline for the simulation study, the IRIS mission parameters (both spacecraft and projected orbit) are used. Table 4 summarizes the relevant parameters of the test orbit.

The IRIS spacecraft is a 3U CubeSat equipped with rate sensors, magnetometer, and sun sensors for attitude estimation, with three-axis torque rods for actuation. The main focus of the study is to compare the attitude knowledge error when using a more advanced sun sensor algorithm. The simulation is initiated to occur during the summer solstice, where the satellite will not be in eclipse at any part of the orbit and simulated for approximately two orbit. The true sun vector is generated using a low-precision formula from the Astronomical Almanac. The magnetometer readings is based on the IGRF-12 model with added white noise modeled from the total RMS noise of the magnetometer (MMC5883MA). The pointing mode is assumed to be sun-pointing throughout the mission operation to ensure that sun sensors yield measurements throughout the simulated period. While perturbations are present, the initial angular velocity is minimum. This ensured that the need for large control torques is not required and allowed the torque rods to keep the satellite in sun-pointing attitude. For this simulation, TRIAD algorithm is used for attitude estimation rather than extended Kalman filter (EKF) or quaternion estimation (QUEST) methods. The TRIAD algorithm offered the simplest deterministic attitude estimation by constructing a third basis using two orthonormal unit vectors based on weighted accuracy of measurement information. More information on TRIAD algorithm can be also found in [16]. The effects of temperature or residual magnetic bias are not considered in the simulation.

## 3. Results and Discussion

The results of the accuracy analysis of the Design 1 (see Table 2) are summarized in Table 5 and highlights the performance of the proposed single-slit DSS design. The 8-bit simulation resulted in a RMSE of 4.7 arcsec, which outperforms linear array based sensors outlined in [17,18,19,20,21,22] and several commercially available sun sensors mentioned in Section 2.4.

Figure 9, Figure 10, Figure 11 and Figure 12 show the estimated versus expected angle outputs from the MCU. As mentioned in Section 2.3, an expected discretization of estimated angles was observed in the data. This phenomenon was most prominent in 1- and 2(1.5)-bit readout modes. This was attributed to the decrease of pixel divisions by the built-in ADC. The increased bit levels of the ADC also contribute to the increased accuracy of the sensor, while the refresh rate is still higher than what most of the COTS components can offer.

The results outlined in Table 5 are further validated using a MATLAB/Simulink attitude estimation simulation. The impact of the different sun sensor readout configurations was examined through the DITL simulation, as described in Section 2.4. Figure 13 below illustrates the sun sensor error during sun-pointing operation using magnetometer and four design options of sun sensor (1-, 2-, 4-, and 8-bit readout operations). The sun sensor reading is compared to the true sun vector generated from propagation, and the difference between the true sun vector and the output of the sun sensor is defined as the sun vector error.

Improvement in sun sensor error can be seen to be proportional to the number of bits used. Sun sensor performance improves significantly from 2- to 4-bit operation as sub-pixel interpolation increased sensor accuracy. The sharp drops in error that can be seen for higher bit operation are indicators of the sun sensors taking advantage of the information provided at higher bit levels to limit the knowledge error. It can be observed that the knowledge error is the same between the different bit levels at different points, such as can be seen around 45 minutes, 70 minutes, and 145 minutes in orbit. After some investigation, it appeared that this is correlated to the incidence angle. As the sun is at a point that is normal to the face, the DSS is able to read at lower bit levels more accurately. This can be attributed through the fact that at full illumination of the relevant area, the amount of information stored in individual bits can be equal at all bit levels. Due to the initial attitude settings, large variation in incidence angle was not tested, however, this may be interesting to investigate in future works as it would cause larger variation in amount of information that can be stored through variation in illumination. Minimal differences in knowledge error can be found between the 1- and 2-bits setting. This can be explained by the photodiode array limitation in 2-bit mode, as previously discussed in Section 2.3. At lower angle differences, it can be observed that the knowledge error is the same between the different bit levels. This is due to lower-illumination levels having the same amount of information even as more bits are used.

In Figure 14, the attitude knowledge is compared during the same maneuver using 4 DSS operation. As magnetometer noise is primarily more dominant compared to the sun sensor noise, marginal improvement is observed from the simulation. No significant improvements from the improved sun sensor reading can be seen, and the level of noise renders any meaningful conclusions as it difficult to tell the different readouts configurations apart. Conversely, by removing the effects of the of the magnetometer it is evident that the proposed DSS shows superior accuracy, especially when reading at 4- and 8-bit levels.

Figure 15 shows a close-up look at the attitude knowledge error in a noise-free setting. The performance correlates with the results of the sun sensor error of the different readout configurations. It is clear here that the 4- and 8-bits readings result in a sub-0.1 deg knowledge error throughout the simulated orbit.

It may be possible that other algorithms which take advantage of higher quality measurements (such as Kalman filter) will have better results, even with the noisy magnetometer readings. The peaks in error can be attributed to the TRIAD algorithm, where, in situations where the sun vector and the magnetic vector have a very similar direction, the accuracy of the algorithm falls. The sun sensor improvement can still be seen when the sun sensor is isolated from other components. This highlights the improvement in performance that can be seen through sub-pixel interpolation. Table 6 summarizes sun sensor error (2-σ) knowledge accuracy for comparison.

The mean sun sensor error significantly improves as the number of bits used increases. The 8-bit DSS improves the mean attitude error by two orders of magnitude relative to the 1-bit DSS. However, the improvements in mean attitude error did not correlate to the improvements of the knowledge error when using the TRIAD attitude estimation method. This can only be attributed to the performance of different estimation methods.

## 4. Conclusions

In this paper, we presented a single-slit DSS design that features low power, computational efficiency, and a very small footprint. Furthermore, a sub-pixel interpolation method was implemented to increase the accuracy of the design by multiple orders of magnitude. Using the developed prototype, simulation study predicted 0.32-degree RMSE when 1-bit readout mode was used, whereas the accuracy improved to better than 0.01-degrees when 8-bit mode is used. Similarly, the attitude mean average error was also reduced using an 8-bit readout mode relative to standard 1-bit readout mode. Lastly, a study was conducted to investigate if the sub-pixel improvements directly translated to improvements in the overall attitude estimation; however, marginal improvement was observed when implemented using TRIAD attitude estimation. Further testing should be considered to improve attitude knowledge accuracy using advanced estimation techniques such as EKF or QUEST. In addition to further investigation on attitude determination algorithm, future research will also focus on improving the DSS calibration technique by using a light source to resent the space-like environment. Current study can also be extended to develop a three-axis attitude determination sensor based on the proposed DSS.

## Figures and Tables

**Figure 1 sensors-21-01472-f001:**
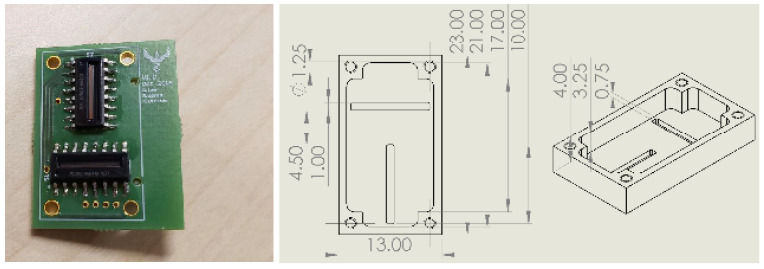
Proposed single-slit digital sun sensor design. Left: installed MLX75306 photodiode, Right: mask design technical drawing.

**Figure 2 sensors-21-01472-f002:**
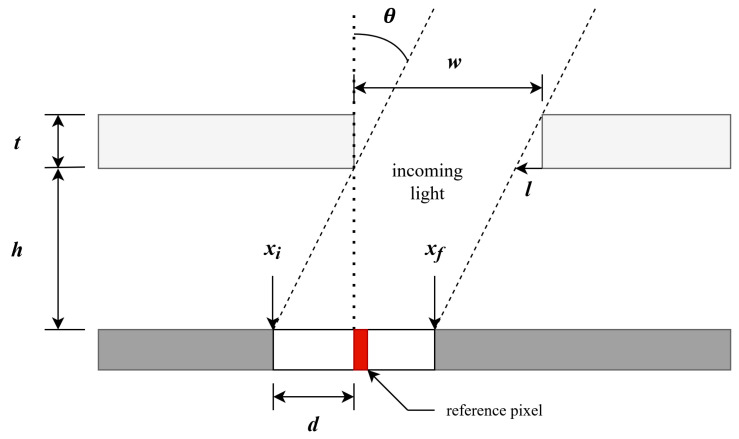
Digital sun sensor readout geometry with single-slit mask. A reference pixel (highlighted in red) is defined as the pixel directly under the mask slit.

**Figure 3 sensors-21-01472-f003:**
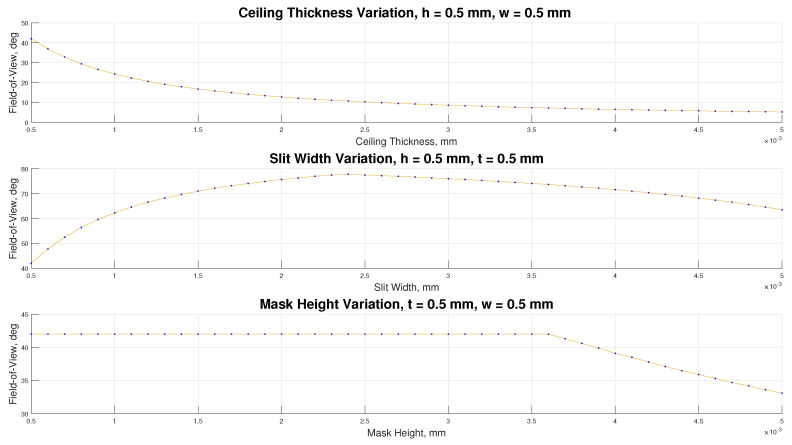
Variations of sun sensor design variables: ceiling thickness, slit width, and mask height.

**Figure 4 sensors-21-01472-f004:**
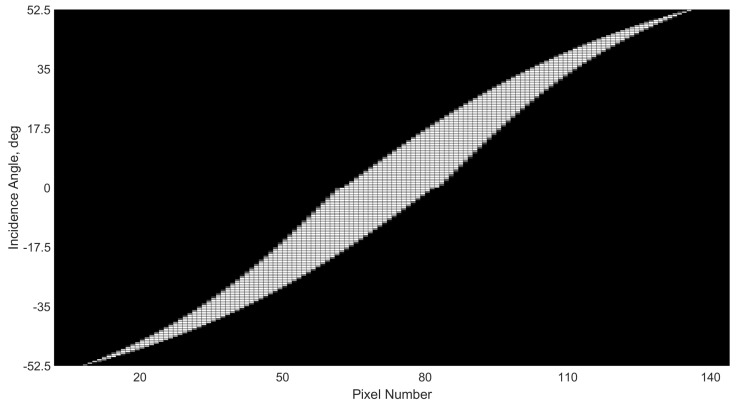
Sweep of expected pixel response at incidence angles ranging from −52.5 through to 52.5 degrees. Illuminated pixels are highlighted in white.

**Figure 5 sensors-21-01472-f005:**
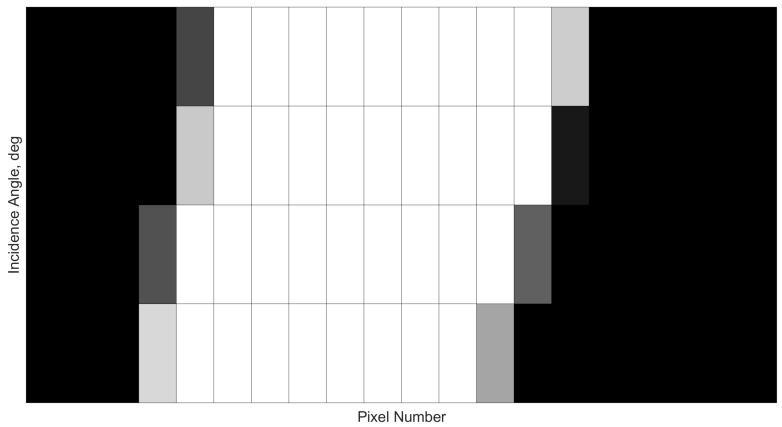
Zoomed-in pixel response to illustrate varying pixel illumination intensity at different incidence angles.

**Figure 6 sensors-21-01472-f006:**
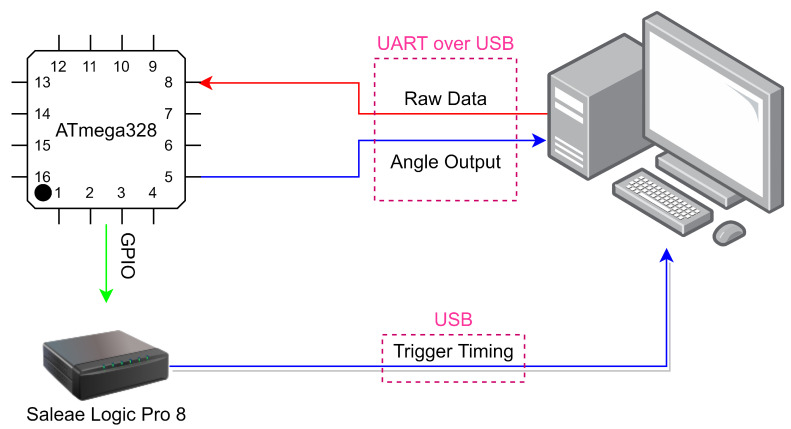
Test configuration 1 system architecture for hardware-in-the-loop simulation test.

**Figure 7 sensors-21-01472-f007:**
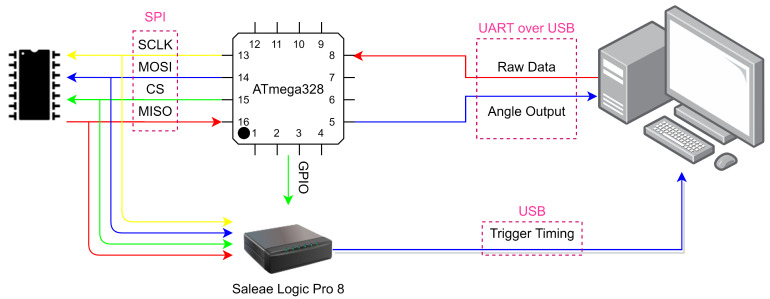
Test configuration 2 system architecture to determine sensor refresh rate.

**Figure 8 sensors-21-01472-f008:**
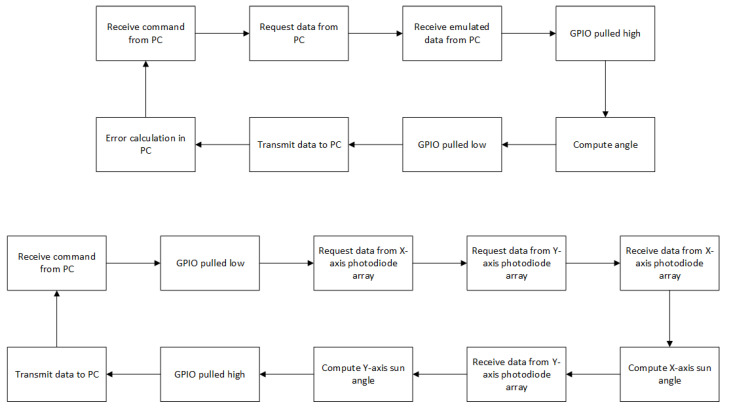
HWIL configurations test flow ((**Top**) Configuration 1, (**Bottom**) Configuration 2).

**Figure 9 sensors-21-01472-f009:**
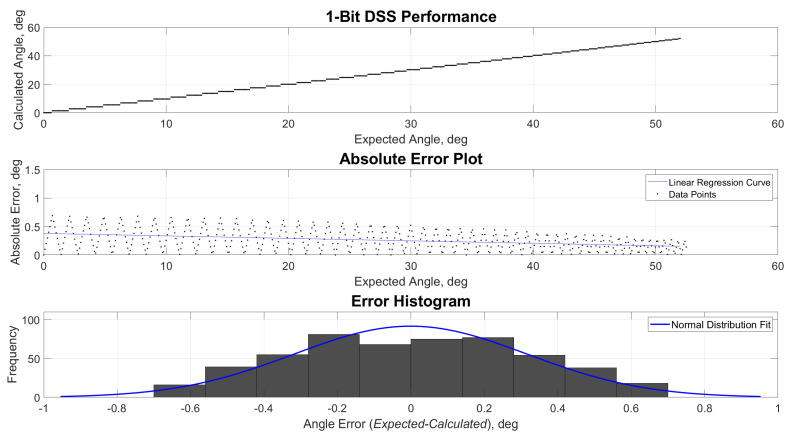
Performance, absolute error and error histogram plots for 1-bit readout mode.

**Figure 10 sensors-21-01472-f010:**
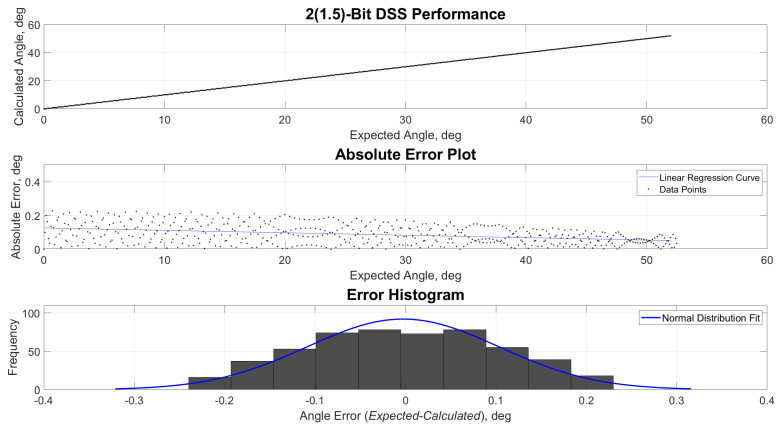
Performance, absolute error and error histogram plots for 2(1.5)-bit readout mode.

**Figure 11 sensors-21-01472-f011:**
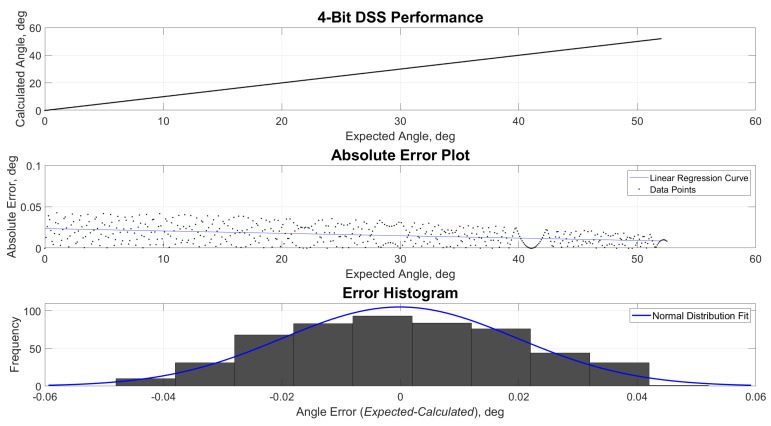
Performance, absolute error and error histogram plots for 4-bit readout mode.

**Figure 12 sensors-21-01472-f012:**
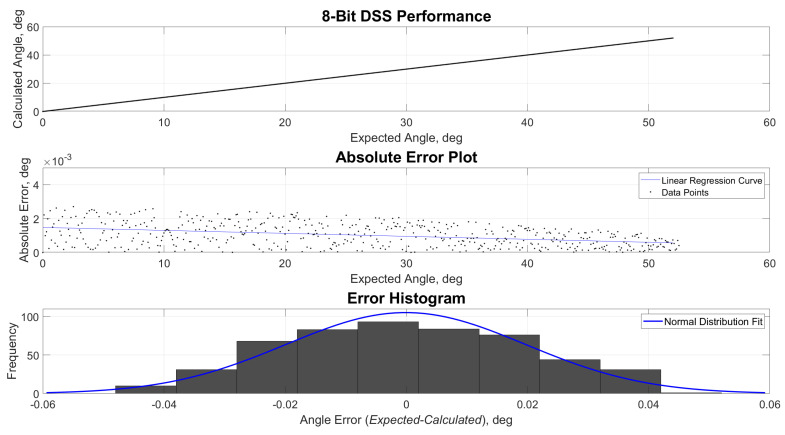
Performance, absolute error and error histogram plots for 8-bit readout mode.

**Figure 13 sensors-21-01472-f013:**
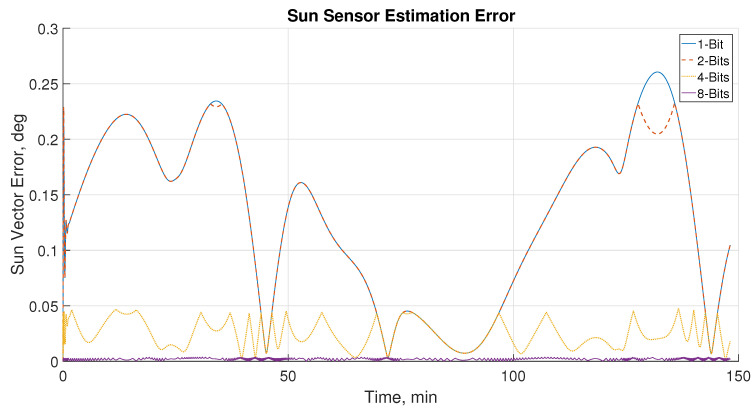
Sun sensor error for the test orbit at 1-, 2-, 4- and 8-bit readout configurations.

**Figure 14 sensors-21-01472-f014:**
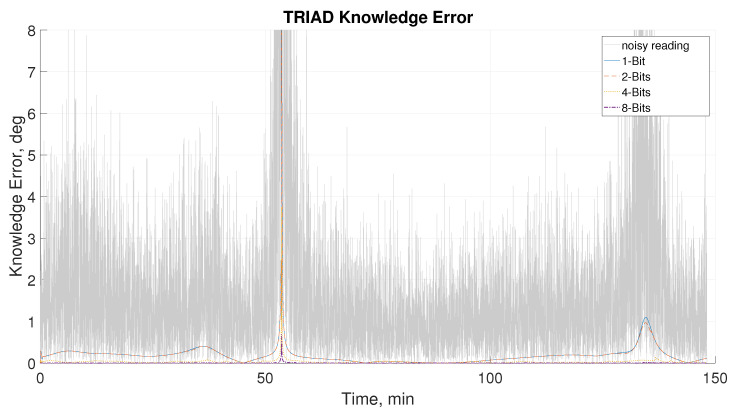
Knowledge error from TRIAD algorithm subjected to noise and knowledge error of different readout configurations with magnetometer noise removed.

**Figure 15 sensors-21-01472-f015:**
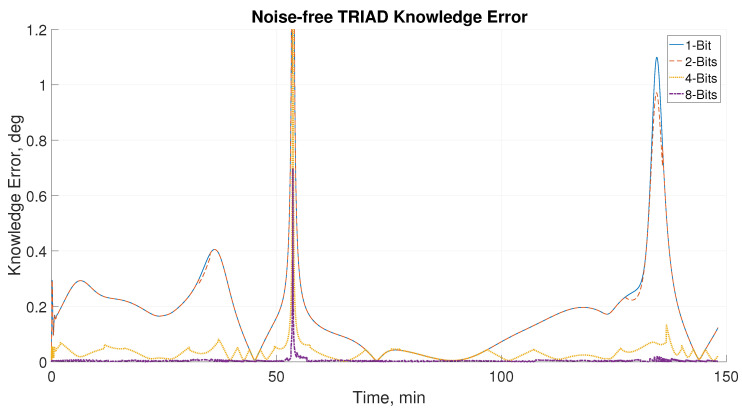
Close-up of knowledge error from TRIAD algorithm of 1-, 2-, 4- and 8-bit readout configurations without noise.

**Table 1 sensors-21-01472-t001:** Digital sun sensor specifications.

Description	Value
Mass	5 g
Field-of-view	105 deg
Update Rate	10 Hz
Accuracy	0.3 deg (1-bit)
	<0.01 deg (8-bit, simulated)
Power	37.5 mW (average)
	130 mW peak
Power Supply	3.3 VDC

**Table 2 sensors-21-01472-t002:** Trade study summary.

Design No.	Ceiling Thickness, *t*, mm	Ceiling Height, *h*, mm	Slit Width, *w*, mm	Field-of-View, FOV, deg	MAE (1-bit), deg
1 (Proposed)	0.75	2.06	1.0	52.5	0.27
2 (Max FOV)	0.5	1.0	1.5	70.0	0.48
3 (Max Accuracy)	0.5	3.60	0.6	41.5	0.17

**Table 3 sensors-21-01472-t003:** Data packet sizes and refresh rates of different output modes.

Readout Modes	Angle Data Length, Bytes	Complete Data Length, Bytes	Total Time, ms	Update Rate, Hz
1-Bit	18	30	6.19	160
2 (1.5)-Bit	36	48	6.51	145
4-Bit	72	83	7.69	127
8-Bit	144	159	10.0	100

**Table 4 sensors-21-01472-t004:** Initial orbital properties for TRIAD simulation.

Description	Value
Date	2019-06-21
Height	700 km
Eccentricity	0
Inclination	97.035 deg
Attitude	Single facet sun exposure
Angular Velocity	0.003 deg/s

**Table 5 sensors-21-01472-t005:** Test configuration 1 accuracy measurements results.

Readout Modes	1-σ Deviation, deg	Avg. Deviation, deg	RMSE, deg
1-bit	0.17	0.91	0.32
2 (1.5)-bit	0.17	0.49	0.16
4-bit	0.01	0.07	0.02
8-bit	<0.01	<0.01	<0.01

**Table 6 sensors-21-01472-t006:** Sun Sensor error summary.

Sun Sensor Type	Mean Attitude Error, deg	Mean Knowledge Error, deg
Coarse (Panel-based)	1–15	0.12–6.0 [15]
1-bit DSS	0.13	1.84
2-bit DSS	0.13	1.85
4-bit DSS	0.025	1.83
8-bit DSS	0.002	1.83

## Data Availability

Not applicable.

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
