# Peer review of "Improved Accuracy of a Single-Slit Digital Sun Sensor Design for CubeSat Application Using Sub-Pixel Interpolation"

_sensors, 2021, doi:10.3390/s21041472_

Round 1

Reviewer 1 Report

This paper investigates the accuracy of a single-slit digital sun sensor, a variation of the sensor designed for the DESCENT CubeSat. The authors define three different geometries for the slit and then investigate the performance of the sensor in function of pixels readout. Results show a good accuracy and low errors in determining the sun direction. In addition, the implementation of the sensor in the attitude determination system indicates that the performance of the reconstruction algorithm (TRIAD) is not directly related to the digital sun sensor one; it is suggest to investigate other estimation techniques that may be advantaged by the sensor accuracy.   

The authors provide a satisfying introduction, referring to the state of the art and to a previous work published on Acta Astronautica; despite similarities, this manuscript presents an independent study worth of publishing.

The following minor suggestions are given to improve the quality of the manuscript:

  • Page 3, line 73. Please check on the journal guidelines if the nomenclature is correct; in case change “3 cm-sq” with “3 cm2”.
  • Figure 2 (left) shows a 5-slits mask, while on the right only a single slit one is drawn. Please check it and uniform or justify the difference.
  • Line 82 indicates that “Key performance specifications of the aperture mask are summarized in Table 1”, but the mentioned table reports the photodiode specifications. I suggest to review the sentence.
  • For sake of completeness, each of the plot titles in figure 4 should report the value of the fixed parameter.
  • Page 5, last line. Is there a reason behind the employment of a non-symmetric simulation range (-52 to +52.5 degrees)?

Author Response

The authors would like to sincerely thank the editors and reviewers for taking the time and effort to read our manuscript and provide valuable guidance. The authors have done their best to address all of the questions and comments raised by the reviewers and have since made major revisions to the manuscript. The manuscript has been re-organized to reflect better readability and flow; removed irrelevant or less useful information; and have added more details to clarify and reinforce certain key points. We have also improved the graphics of the paper and added new plots to illustrate key findings.

Note: In the following, reviewer’s comments are in italic, while author responses are in regular font. References to specific lines are given when applicable.

Reviewer 1 Comments

  1. Page 3, line Please check on the journal guidelines if the nomenclature is correct; in case change “3 cm-sq” with “3 cm2”.

The nomenclature has been corrected, and the corrected line can be seen in Line 77.

  1. Figure 2 (left) shows a 5-slits mask, while on the right only a single-slit one is Please check it and uniform or justify the slit one is drawn. Please check it and uniform or justify the difference.

The figure has been updated to correctly reflect the consistency of the paper.

  1. Line 82 indicates that “Key performance specifications of the aperture mask are summarized in Table 1”, but the mentioned table reports the photodiode I suggest to review the sentence

“aperture mask” has been replaced with “sun sensor” and can be seen in Line 89. The labels for Table 1 has been updated to indicate that the variables are for the sun sensor as a whole, and not just the aperture. The corrected table can be seen below Figure 1, after Line 89.

  1. For sake of completeness, each of the plot titles in Figure 4 should report the value of the fixed

The authors’ agree to the suggestion. We have modified the figure so that the constant values used are now shown.

  1. Page 5, last line. Is there a reason behind the employment of a non-symmetric simulation range (-52 to +52.5 degrees)?

This is a typo. The fixed range of -52.5 to +52.5 degrees can be seen in Line 132.

Reviewer 2 Report

The paper introduces a two-slit sun sensor design together with an angle interpolation method to determine a spacecraft’s attitude relative to the Sun. The article falls short of an acceptable publication standard in multiple aspects which are outlined below. I recommend not to publish this article and to ask the authors to re-write the article for re-submission. Before re-submission, I urge the authors to make the text more concise and descriptive. What information is important and what is not? The general structure of the text needs to be re-worked and relevant information needs to be placed in focused sections. Figures show irrelevant information and are redundant. Re-think what information is worth showing and how the Figures can be more intuitive.

General comment:

The general article structure and content is lacking to the point that I cannot identify key developments that are mentioned in the article’s title.

All Figures require more attention. Captions need to be more descriptive. Labels and numbers need to be larger to be readable. Units are missing.

It is not clear why Figure 1 is important. How does this Figure relate to the rest of the article?

Section 2.2 describes the trade study for the sun sensor mask. Is this essential work? If so, it should receive more treatment than half a page.

Line 104: You don’t need to set variables constant in a multi-variable analysis. Depending on the technique all variable can be varied at the same time. The goals behind the optimization and the optimization technique require much more description.

The first sentence of section 2.3 does not make sense.

Line 129: Why not use grey-scale values?

I assume that the interpolation algorithm is a key part of this analysis. After reading the description (section 2.3), it is not clear to me how it works in conjunction with the Figure showing the illuminated pixels as a function of incidence angle.

Line 138: First you talk about the conversion of analog to digital using varying bit values. Here you suddenly switch to talk about resolution errors. Should this be a new section? How is this related to bit values?

Line 143: You write x%. Is there a number missing here?

Line 154: typo “the the”

Why is Table 2 in the result section and not in the sensor trade study section?

Figure 10: It would be more instructive to show the error between expected and measured value.

Author Response

Response to Reviewer Comments

The authors would like to sincerely thank the editors and reviewers for taking the time and effort to read our manuscript and provide valuable guidance. The authors have done their best to address all of the questions and comments raised by the reviewers and have since made major revisions to the manuscript. The manuscript has been re-organized to reflect better readability and flow; removed irrelevant or less useful information; and have added more details to clarify and reinforce certain key points. We have also improved the graphics of the paper and added new plots to illustrate key findings.

Note: In the following, reviewer’s comments are in italic, while author responses are in regular font. References to specific lines are given when applicable.

Reviewer 2 Comments

  1. Before re-submission, I urge the authors to make the text more concise and descriptive. What information is important and what is not? The general structure of the text needs to be re-worked and relevant information needs to be placed in focused Figures show irrelevant information and are redundant. Re-think what information is worth showing and how the Figures can be more intuitive.

Structure of the text has gone through a revision, and will be discussed with responses to comments listed below. Redundant text and figures have been removed: Figures 1, 10, and 14 have been removed. Figure 8 has been added to clarify the hardware-in-the-loop workflow. Specific edits will be pointed out through responses to the comments and questions listed below.

  1. The general article structure and content is lacking to the point that I cannot identify key developments that are mentioned in the article’s

The article structure has been revised, and content has been reorganized with redundant infor- mation removed in order to improve the clarity of the paper. Specific edits will be pointed out through responses to the comments and questions listed below.

  1. All Figures require more Captions need to be more descriptive.  Labels  and num-  bers need to be larger to be readable. Units are missing

All the figures have been updated to improve clarity and readability. Specifically, Figures 1, 10, and 14 has been removed to improve focus of the paper. Figure 10 has been replaced with Figures 9-12 to show data in a meaningful manner. Figure 2 have been added with parameters which are discussed in the paper. Figures 12 and 13 (now 14 and 15) have been replaced with graphs showing a clearer image of what has been examined. Specific changes and their reasoning

are discussed with the responses specific to the figures below.

  1. It is not clear why Figure 1 is important. How does this Figure relate to the rest of the ar-  ticle?

The authors wanted to point some original works in the literature and their developments. We decided that the figure was not relevant and have since pointed the reader to the original reference. The figure has been removed in the revised manuscript.

  1. Section 2.2 describes the trade study for the sun sensor mask. Is this essential work? If so, it should receive more treatment than half a

The feedback has given the opportunity for a critical review of the section. The trade study was deemed to dilute the focus of the paper and has been removed. The entirety of Section 2.2 have been changed to discuss the key mechanical parameters. The name of Section 2.2 have been changed from “Mask Design Trade-study” to “Mask Mechanical Design”

  1. Line 104: You don’t  need  to set variables constant in a multi-variable analysis.  Depending  on the technique all variable can be varied at the same time. The goals behind the optimization and the optimization technique require much more description

The authors are not familiar with the technique the reviewer is referring to here. The manuscript’s analysis did not vary all the variables at the same time in retrospect to maintaining the integrity of the paper. However, we have drastically improved this section to include more description on the technique used and the apparent goals.

  1. The first sentence of section 3 does not make sense.

The beginning of Section 2.3 have been rewritten, discussing previous work and the reasoning behind the sub-pixel interpolation. Thank you.

  1. Line 129: Why not use grey-scale values?

The RGB values were used to represent grey values. The RGB values were varied between [0,0,0] and [1,1,1], where a 10% illuminated pixel would have the RGB values of [0.1,0.1,0.1]. This example was used in the text the illustrate the colors more clearly in the text, and can be seen in Lines 135-136.

  1. I assume that the interpolation algorithm is a key part of this analysis. After reading the description (section 2.3), it is not clear tome how it works in conjunction with the Figure showing the illuminated pixels as a function of incidence

The interpolation description was significantly expanded and was supplemented with algebraic ex- amples to show the step-by-step interpolation process per reviewers comments. The authors hope that the work in section 2.3 is more clear. The additional paragraph detailing more information is shown in Lines 156-167.

  1. Line 138: First you talk about the conversion of analog to digital using varying bit values. Here you suddenly switch to talk about resolution errors. Should this be a new section? How is this related to bit values?

Thank you for bringing attention to a possible point of confusion. The analog-to-digital conversion causes discretization error to be introduced, since the illumination ratio cannot be perfectly read using the ADCs (Analog-to-Digital Converter). In the 8-bit case, if the ADC conversion was done without errors, a 65% illuminated pixel would return 165.75. However, since the outputs must be integers, the hardware would round this value to 166, therefore introducing a 0.098% error. The section was expanded and supplemented with example cases to make the process more clear for the reader per reviewers comments. The previous text which were in Lines 138-150 has been removed, and the new text that has replaced it can be seen in Lines 145-155.

  1. Line 143: You write x%. Is there a number missing here?

The line has been removed entirely with the correction made in response to Comment 10.

  1. Line 154: typo “the the”

The typo has been fixed. Thank you. The corrected line can be seen in Line 171

  1. Why is Table 2 in the result section and not in the sensor trade study section?

Table 2 was relocated to Section 2.2 to address the organization issue raised. It was also noted that the chart suffices for the trade study details, and the text describing the trade study in detail has been removed to avoid detracting from the focus of the paper and has been replaced by pa- rameter details. Section 2.2 has been changed from mask design trade-study to mask mechanical design as per the response to Comment 5.

  1. Figure 10: It would be more instructive to  show  the  error  between  expected  and  measured value

Figure 10 was removed and replaced with figures showing both the performance, absolute er- ror and the error histogram. The new Figures 9-12 can be seen in Section 3.

Reviewer 3 Report

This paper deals with the development of a sun-sensors for small satellites. Attitude determination of small satellites is an interesting topic and the improvement of the sensors’ technology is of interest for the scientific community.

The presented work has an average level of novelty and seems promising in terms of developed solution. However. the organization of the paper must be improved, statements are sometimes too generic and ambiguous, and the description of the results requires supplementary discussion. These issues make the paper hard to be read.

I can support the publication of this paper only after a major revision that addresses these points:

  • A better organization of the paper, in particular authors should improve the link between the test description and setup in section 2 and the results in section 3. The reader struggles to associate the results. Please implement solutions that improve the readability: for example, adding subtitles or labels to the different tests
  • Section 2.2: the description is to generic, please consider adding a further description of the trade-study or add references that could help the reader to understand the trade-process in detail. In particular, how the optimized solution can be deduced from the fig 4.
  • Line 140-145: what are “the specifications”? Please provide a more detailed explanation and/or add the datasheet(s). Are these specifications an issue only for the case of 2-bits mode?
  • Line 173 and Fig 8: what is the MLX75306 IC? why is it included in the loop through the SPI bus? It seems that it should emulate the sun sensors behaviours. If it is, why is it not commanded by the main PC? Improve the description of the setup.
  • Line 182-196 the description of this important step of validation should be improved. As far as I understand, authors substitute the output of the sun sensors of the IRIS with the output of the proposed sun sensor performing a sort of “day-in-the -life” simulation. If it is, please reword this part, describing the simulator architecture. Finally, authors refer to IRIS mission without providing reference.
  • Line 203-205: this statement is very questionable without clarification. What is the “particular application”? Moreover, the authors should clarify what are “the most CubeSat and NanoSat missions” and what does it characterize these missions. What are the reference missions that the authors have in mind and that do not require a fine pointing?
  • Line 207-212: I think that it is a very important result that would allow to compare the proposed solution with the performance of the star trackers. However, it should be clarified that the actual performance of the proposed sensor is not validated in a relevant environment. Then performance of proposed solution will be affected by EMI, hard thermal environment, radiations and so on, that reduces the overall performance. Definitely, I think that the comparison could be weak without clarification.
  • Fig 10: I suppose that x label should be “estimated angle”. The scale and readability of the figures do not allow to appreciate the comparison. Moreover, why the range of comparison is about 37° - 90°? Please improve the figures quality or identify a better solution to present these results.
  • Results of table 4 require further description. Better, in section 2, when author describe the test, it should be clarified what are the parameters under analysis.
  • The results on fig 11 seem promising but, as said before, the description of the test could help to better understand these results. It means that authors should better clarify the mission profile, the expected output and the actual ones.
  • Fig 12 and Fig 13 is not easily readable, authors could find a better way to show the results, for example, a subfigure for any set of results (i.e. 1bit, 2 bits, 4bits, 8bits) could be included.
  • Line 259-268. Since TRIAD algorithm is a very approximate method to determine the attitude of a satellite and, for sure, the obtained results from the proposed solution would be better appreciated and useful when KF is adopted, I think that, in the framework of the paper, authors should stress that the obtained results validate the proposed solution.
  • Conclusion are brief and trivial. Could the author improve them, providing a roadmap of future works. Do the authors plan to perform test in relevant conditions? Are the authors considering including the sensors in the loop, using the Sun or a Sun simulator as “stimulating” ground support equipment?

Other minor improvements/revisions of the paper can be:

  • Add reference in line 62 for DTUsats missions
  • Add reference in line 94 for DESCENT
  • Please confirm the interval at the end of pag 5

Author Response

Response to Reviewer Comments

The authors would like to sincerely thank the editors and reviewers for taking the time and effort to read our manuscript and provide valuable guidance. The authors have done their best to address all of the questions and comments raised by the reviewers and have since made major revisions to the manuscript. The manuscript has been re-organized to reflect better readability and flow; removed irrelevant or less useful information; and have added more details to clarify and reinforce certain key points. We have also improved the graphics of the paper and added new plots to illustrate key findings.

Note: In the following, reviewer’s comments are in italic, while author responses are in regular font. References to specific lines are given when applicable.

Reviewer 3 Comments

  1. A better organization of the paper, in particular authors should improve the link between the test description and setup in Section 2 and the results in Section The reader struggles to associate the results. Please implement solutions that improve the readability: for example, adding subtitles or labels to the different tests

Thank you for your comments. The paper has been restructured to address the issues raised. Specific edits will be pointed out through responses to the comments and questions listed below.

  1. Section 2: the description is to generic, please consider adding a further description of the trade-study or add references that could help the reader to understand the trade-process in detail. In particular, how the optimized solution can be deduced from the fig4

Considering the reviewer’s suggestion regarding the trade-study (formerly Section 2.2), we have removed the irrelevant information on the comparative analysis; instead, we described the final design parameters that were used for the proposed study. In providing the details, we discussed the important parameters that we considered, namely mask height, mask thickness and slit width, and the resulting performance of the sun sensor derived from the design.

  1. Line 140-145: what are “the specifications”? Please provide a more detailed explanation and/or add the datasheet(s). Are these specifications an issue only for the case of 2-bits mode?

Added the “specification”, which is the average byte. This value, along with CRC, is used to check and correct for readout errors from the photodiode arrays. New text which explains this can be seen with the new paragraph in Lines 156-167

  1. Line 173 and Fig 8: what is the MLX75306 IC? why is it included in the loop through the SPI bus? It seems that it should emulate the sun sensors If it is, why is it not com- manded by the main PC? Improve the description of the setup.

The MLX75306 IC is mentioned in Section 2.1, and is the photodiode array used in the DSS. It was included in the second configuration of the hardware-in-the-loop test to account for the data acquisition time of the IC. The Hardware-in-the-Loop description was expanded and the setup was clarified. The new description can be seen in Lines 172-176, and the new text which describes the setup can be seen in Lines 183-191, 203-211 which is supported by the newly added Figure 8.

  1. Line 182-196: the description of this important step of validation should be As far as I understand, authors substitute the output of the sun sensors of the IRIS with the output of the proposed sun sensor performing a sort of “day-in-the-life” simulation. If it is, please reword this part, describing the simulator architecture.

Some of the description of this step was placed in section 3; this has now been moved into the proper section which we hope clarifies the simulation architecture. The changes have been applied near the end of Section 2.4, and the new text which describes the setup can be seen between Lines 223-251. Table 4 has also been moved with the text.

  1. Finally, the authors point to the IRIS mission without providing

IRIS as previously named as ManitobaSat-1. Reference 12 points to this mission.

  1. Line 203-205: this statement is very questionable without What is the “partic- ular application”? Moreover, the authors should clarify what are “the most CubeSat and NanoSat missions” and what does it characterize these missions. What are the reference missions that the authors have in mind and that do not require a fine pointing?

The authors agree with the reviewer in this comment. We realized that the statements were too generic and ambiguous. We have since improved the manuscript overall and removed any claims any mission in general.

  1. Line 207-212: I think that it is a very important result that would allow to compare the pro- posed solution with the performance of the star However, it should be clarified that the actual performance of the proposed sensor is not validated in a relevant environment. Then perfor- mance of proposed solution will be affected by EMI, hard thermal environment, radiations and so on,

The authors would like to thank the reviewer here for pointing this out. The study conducted in the manuscript did not validate the results in a relevant environment and most, certainly all factors mention would play a detrimental role to the performance. We have tried to replicate the environment through HWIL using simplified models and concepts. We have revised the manuscript to ensured that this was properly conveyed.

  1. Fig 10: I suppose that x label should be “estimated angle”. The scale and readability of the figures do not allow to appreciate the Moreover, why the range of comparison is about

37° -90°? Please improve the figures quality or identify a better solution to present these results.

Figure 10 was replaced with 4 separate plots, Figures 9-12. Each shows the performance, ab- solute error and error histograms. X-axis was changed to be “Estimated Angle”. Font size was increased, and figure size overall was increased for added readability. The range of comparison was showing the complementary angle, which has been corrected to show the range of 0 to 52.5 degrees. Since sun sensor behavior is uniform on [-52,5,0] and [0, 52.5] degree ranges, only one plot is included.

  1. Results of table 4 require further Better, in section 2,when author describe the test, it should be clarified what are the parameters under analysis.

Table 4 was moved to Section 2.3 and the section was expanded and additional clarification was added, as per the response to Comment 5.

  1. The results on fig 11 seem promising but, as said before, the description of the test could help to better understand these It means that authors should better clarify the mission pro- file, the expected output and the actual ones.

Added justifications for the simulation properties. The reasoning behind the simulation is added in Lines 229-233, to make clear that is the result that will be used to validate the sun sensor per- formance. Clarified the mission profile and satellite properties in the new text placed in Section

2.4. as per the response to Comment 5. Line 247-248 stating “If the sun was at a point that was normal to the x-face, the sun sensor was able to read that even at lower bit levels” has been replaced by Lines 277-278 stating “As the sun is at a point that is normal to the face, the DSS is able to read at lower bit levels more accuratel” to clarify the situation.

  1. Fig 12 and Fig 13 is not easily readable, authors could find a better way to show the re- sults, for example, a subfigure for any set of results (i.e. 1bit, 2 bits, 4bits, 8bits) could be

Figure 12 and 13, now Figure 14 and 15, has been adjusted to emphasize the improvement due to the sub-pixel interpolation. The noisy graph did not carry much information, and has been replaced with a figure which shows the noisy reading in the background with the noise-free readings. A close-up of the noise-free reading is used to improve readability between each readout configura- tion. Figure 14 has been deemed redundant for driving home the message of an improved sensor, and has been removed.

  1. Line 259-268. Since TRIAD algorithm is a very approximate method to determine the at- titude of a satellite and, for sure, the obtained results from the proposed solution would be better appreciated and useful when KF is adopted, I think that, in the framework of the paper, authors should stress that the obtained results validate the proposed

Added text which directly addresses Figure 12, now Figure 14, in order to stress the improve- ment that can be seen in knowledge error using the proposed solution. Lines 297-298 and Lines 303-305 have been added to emphasize the validation of the sun sensor. The validation is also shown through the attitude error values shown in Table 6.

  1. Conclusion are brief and trivial. Could the author improve them, providing a road map of future Do the authors plan to perform test in relevant conditions? Are the authors consid- ering including the sensors in the loop, using the Sun or a Sun simulator as “stimulating” ground support equipment?

Added the “specification” as seen in paragraph starting from Line 179, which is the average byte. This value, along with CRC, is used to check and correct for readout errors from the photodiode arrays.

  1. Add references to line 62 for DTUsats missions and line 94 for DESCENT

We have added Reference 11 and Reference 13.

  1. Please confirm the interval at the end of page 5

The interval for the sweep is 0.5 deg. This information is added in the manuscript.

Round 2

Reviewer 2 Report

The authors provided significant improvements since the last version.

  • Open points include font size of the figures is generally too small and should be enlarged for better readability in the final version.
  • Figures should be able to stand alone with captions. Meaning that just by looking at the Figure and reading the caption, a reader should be able to grasp the points made by a figure. That includes description of what is being shown and one or two significant observations. The captions would benefit from being extended in that fashion.

Author Response

The authors would like to sincerely thank the reviewers for their time and guidance. The figure quality has been improved as per the suggestions and we have added more description to the captions. 

Reviewer 3 Report

This new version of the paper is fine. Authors accepted the suggestions and mainly implemented them. 

I do agree with the publication of this paper in the present form.

Author Response

The authors would like to sincerely thank the reviewers for their time and guidance